# Bacteriocin Occurrence and Activity in *Escherichia coli* Isolated from Bovines and Wastewater

**DOI:** 10.3390/toxins11080475

**Published:** 2019-08-15

**Authors:** Andrew Cameron, Rahat Zaheer, Emelia H. Adator, Ruth Barbieri, Tim Reuter, Tim A. McAllister

**Affiliations:** 1Faculty of Veterinary Medicine, University of Calgary, Calgary, AB T2N 4N1, Canada; 2Lethbridge Research and Development Centre, Lethbridge, AB T1J 4B1, Canada; 3Department of Food Science and Human Nutritional Sciences, University of Manitoba, Winnipeg, MB R3T 2N2, Canada; 4Alberta Agriculture and Forestry, Lethbridge, AB T1J 4V6, Canada

**Keywords:** *Escherichia coli*, bacteriocin, colicin, microcin, antimicrobial peptide, antimicrobial resistance, cattle, wastewater, STEC

## Abstract

The increasing prevalence of antimicrobial resistant (AMR) *E. coli* and related Enterobacteriaceae is a serious problem necessitating new mitigation strategies and antimicrobial agents. Bacteriocins, functionally diverse toxins produced by most microbes, have long been studied for their antimicrobial potential. Bacteriocins have once again received attention for their role as probiotic traits that could mitigate pathogen burden and AMR bacteria in livestock. Here, bacteriocins were identified by activity screening and whole-genome sequencing of bacteriocin-producers capable of inhibiting bovine and wastewater *E. coli* isolates enriched for resistance to cephalosporins. Producers were tested for activity against shiga toxin-producing *E. coli* (STEC), AMR *E. coli*, and related enteric pathogens. Multiple bacteriocins were found in 14 out of 90 *E. coli* isolates tested. Based on alignment within BACTIBASE, colicins M, B, R, Ia, Ib, S4, E1, E2, and microcins V, J25, and H47, encoded by identical, variant, or truncated genes were identified. Although some bacteriocin-producers exhibited activity against AMR and STEC *E. coli* in agar-based assays, most did not. Despite this idiosyncrasy, liquid co-cultures of all bacteriocinogenic isolates with luciferase-expressing generic (K12) or STEC *E. coli* (EDL933) resulted in inhibited growth or reduced viability. These abundant toxins may have real potential as next-generation control strategies in livestock production systems but separating the bacteriocin from its immunity gene may be necessary for such a strategy to be effective.

## 1. Introduction

Bacteriocins produced by the Gram-negative Enterobacteriaceae *Escherichia coli* are ribosome-synthesized toxins known as colicins and microcins [1]. Colicins are typically large proteins of high molecular mass (40–80 kDa), whereas microcins are low molecular weight (<10 kDa) (poly)peptides [2,3]. Often produced exclusively under stress, such as the SOS response induced by mitomycin C [4], both colicins and microcins are capable of killing a narrow spectrum of competing *E. coli* and phylogenetically related bacteria [2]. A wide array of colicins and microcins have evolved numerous cytotoxic mechanisms, including include pore formation; degradation of peptidoglycan precursors; phosphatase activity; RNAse activity (often targeting 16S rRNA and specific tRNAs); and DNAse activity [2]. Producers are protected from self-killing by co-synthesizing a specific immunity protein which protects against the action of the bacteriocin through various mechanisms. Some *E. coli* solely produce the immunity factor to gain a competitive advantage against producers [5]. Colicins and microcins intrude into susceptible *E. coli* by exploiting conserved transport (i.e., frequently iron uptake systems [6]), diffusion, or efflux systems as specific receptors. The narrow target range of colicins is likely due to the requirement for specific outer membrane receptors on the cell surface. The method of colicin uptake in the inner membrane has given rise to a system of classification: In general, Group A colicins have parasitized the Tol system for translocation into *E. coli* and Group B colicins exploit Ton for translocation [2,7]. Indeed, the ‘Tol’ system is named for the fact that Tol mutants are ‘tolerant’ to colicins. Microcins may also be categorized into two groups depending on whether or not the microcin is post-translationally modified during maturation of the microcin peptide into the active form [3]. Group A and B colicins also differ in the way the colicin is released from the producer *E. coli*: Group A colicins typically encode a lysis protein thought to ensure the efficient and maximal release of colicin into the environmental milieu. Consequently, the release of Group A colicins is lethal to the producer *E. coli* cell [8]. Colicins have a functionally distinct modular domain structure: A translocation domain (N-terminal), a receptor-binding domain, and toxic domain (C-terminal) [2]. Some microcins have a modular design, featuring a toxic domain (N-terminal) and an uptake domain (C-terminal) [9]. These designs have likely facilitated the evolution of diverse toxicities via recombination.

Following nearly a century of extensive research, a great deal is known about the complexities of the molecular biology of *E. coli* bacteriocin function, immunity, and regulation [2]. Bacteriocin production has traditionally been an important factor in the selection of novel probiotic strains [10]. Although inhibition of pathogenic *E. coli* by bacteriocin-producers has been well-documented in vitro [2,11], their efficacy in vivo, as produced by live probiotics, has been less than stellar [10,12,13]. Relatively few studies have definitively characterized the ability of bacteriocin-producers to colonize and benefit the host: recent work has shown probiotic gastrointestinal colonization and efficacy remains controversial, transient, host-individualized, and limited [14]. Despite this, the administration of bacteriocin-producers *in lieu* of purified bacteriocins is thought to be a more effective approach than treatment with purified products [15]. To date, one of the only purified bacteriocins routinely used is nisin, a peptidyl bacteriocin discovered in the 1920s in the Gram-positive *Lactococcus lactis* subsp. *lactis*, which today is in large-scale usage as a food preservative [15]. Notwithstanding bacteriocins naturally produced in fermented food products, the exploitation of the plethora of known bacteriocins remains limited. However, as antimicrobial resistance (AMR) spreads and more countries limit the use of traditional antimicrobials, the need to develop alternative antimicrobials and pathogen mitigation strategies will grow. This requires the identification of novel bacteriocins, as well as rapid screening methodologies to search for putative bacteriocin-producers. Critically, it will be important to establish efficacy against pathogens likely to be already resistant to key antimicrobials, such as the β-lactam class drugs.

The first colicin was identified by Gratia in 1925, who called it ‘principle V’, a heat-labile microcin present in cultures of *E. coli* V (for ‘virulent’) now known most frequently as colicin V. Nearly a century later, it is evident that most bacterial species produce bacteriocins and nearly all living organisms produce antimicrobial peptides (AMPs) [3]. Bacteriocins have long been thought to be important elements in bacterial ecology and have been linked to both probiotic effects and virulence in *E. coli* [16,17]. The potential for bacteriocin-producing probiotics to serve as alternatives to antimicrobials, particularly in food animal species, has received increased attention in an era defined by increasing AMR [17,18]. Of concern for Enterobacteriaceae are extended-spectrum β-lactamases (ESBLs), which confer resistance to most β-lactam antimicrobials, including penicillins and cephalosporins. Human infections with ESBL-producing bacteria are associated with poor clinical outcomes [19,20] and human illness due to infections with shiga toxin-producing *E. coli* (STEC) is a global health concern [21]. The development of bacteriocins or bacteriocinogenic probiotics capable of lowering the burden of AMR *E. coli* or excluding STEC in food animals could prevent zoonotic transmission of AMR *E. coli* and STEC. Fecal shedding of STEC O157:H7 and other serotypes in feedlot cattle is common [22]. Although probiotic colicinogenic *E. coli* strains have been found to exhibit activity against STEC in vitro [11], many probiotics have shown mixed results in reducing O157:H7 fecal shedding in cattle [12]. Moreover, bacteriocins produced by Gram-positive bacteria are not typically active against Gram-negative bacteria. Therefore, it is rational to conclude that Gram-negative bacteriocin-producers offer greater potential as probiotics against *E. coli*. The aim of this study was to identify putative bacteriocins capable of inhibiting both AMR *E. coli* and STEC. Given the increasing prevalence of AMR *E. coli*, we reasoned those same bacteria might harbor novel bacteriocins active against competing AMR *E. coli*. Here, by screening for evidence of the production of diffusible substances with anti-*E. coli* activity, we characterized the bacteriocin content of a library of *E. coli* isolated from bovine feces and wastewater via whole-genome sequencing.

## 2. Results

### 2.1. Functional Screening and Genomic Identification of E. coli Bacteriocins

A library of 90 *E. coli* (derived from bovine feces, feedlot-associated catchbasins, and wastewater influent) was treated with mitomycin C and then tested for antimicrobial activity against *E. coli* K-12 strain MG1655 and the STEC O157:H7 strain EDL933. Of these, 14/90 isolates were observed to produce inhibition halos (i.e., diffusible substances) in lawns of MG1655, whereas only 4/14 produced halos in EDL933 lawns (Figure 1A). To confirm that inhibition was not due to phage activity in these isolates, serial dilutions of culture supernatants were spotted onto lawns of MG1655 and phage plaques were confirmed absent (not shown). These isolates, deemed putative ‘bacteriocin-producers’, were sequenced and their associated bacteriocins were identified bioinformatically by *blastx* alignment to BACTIBASE, a database of bacterial antimicrobial peptides. Bacteriocin genes were identified in all isolates, with some harboring as many as four unique bacteriocins (Table 1). Of those identified, the most abundant were colicins identical or near identical to colicins B and M, which were found in up to 7/14 putative bacteriocin-producer *E. coli*. The next most common bacteriocins were close homologues or identical matches to colicin (microcin) V (found in 5/14 isolates), and colicins Ia and Ib, which were each found in 2/14 isolates. Less frequent were the microcins H47 and J25 and colicins R, E1, E2, and S4.

### 2.2. Bacteriocins Identified, Genomic Context, and Similarity to Known Colicins and Microcins

Except for microcin H47, all the bacteriocins identified were found on contigs aligning to plasmid sequences in the NCBI *nr* database. Interestingly, several contigs aligned most closely (>95% nucleotide identity; highest query coverage) with plasmids found in various *Salmonella enterica* isolates (Table 1). However, individually, each bacteriocin aligned most closely with sequences from *E. coli* and many had identical amino acid (*aa*) sequences to their prototype bacteriocins (Figure 2). 

#### 2.2.1. Colicins B and M

Colicins B and M are ‘Group B’ colicins encoded by *cba* (colicin B activity gene) and *cma* (colicin M activity gene). The gene operons for colicins B and M are co-occurrent at frequencies greater than those expected by chance [23]. Here, each colicin was invariably found transcribed in the same direction and was adjacent to the cognate immunity genes, *cbi* (colicin B immunity protein) and *cmi* (colicin M immunity protein) (Figure 3). Without exception, the *cbi* and *cmi* immunity genes were found transcribed in the opposite direction to *cba* and *cma*. Consistent with the literature [2], no lysis proteins were associated with either colicin B or M, which otherwise were co-located on contigs containing genes involved in conjugal plasmid transfer or plasmid maintenance (e.g., *tra* genes, encoding conjugation proteins; *spo0J*, chromosome-partitioning protein; and *parM*, plasmid segregation protein). Colicins B and M were frequently associated with recombinases (e.g., *xerD*, tyrosine recombinase) and transposases. Consistent with this, contig BLASTs with the NCBI *nr* database indicated the contigs containing colicin B and M had highest identity with self-transmissible plasmids originating either in *E. coli* or in *Salmonella enterica* (Table 1). Thus, the contig sequences containing colicin B and M described here are likely plasmid-borne. In isolates 0043M and 0012I, the contigs encoding colicin B and M carried aminoglycoside (*aph(6)-Id*) and tetracycline (*tet(C)*) resistance genes. For isolate 0012I, the colicin-bearing plasmid is likely sufficient for the antimicrobial resistance phenotype (streptomycin; oxytetracycline) of the isolate.

A pore-forming colicin, Colicin B kills sensitive cells by forming ion channels that depolarize the cytoplasmic membrane. Structurally, colicin B does not have clearly delineated receptor-binding and translocation domains: The N-terminal (290 *aa*) contains both domains and the C-terminal contains the pore-forming domain. In isolates 0638J and 0315J, the *cba* (full length: 511 *aa*) product was truncated with a premature termination codon into two putative fragments: The first fragment (94 *aa*) had 49.1% *aa* identity to the N-terminal of the *cba* prototype (UniProtKB: P08520) and the second (142 *aa*) had 93.7% *aa* identity to the C-terminal. The cognate immunity genes for the truncated *cba* shared 93.7–94.9% *aa* identity to the *cbi* prototype (UniProtKB: P22426) vs. 99.8–100% *aa* identity for the full-length *cba* genes. Likewise, the full-length colicin M *cma* identified here encoded for 271 *aa* proteins, except for isolates 0638J and 0315J, where the protein was truncated with a premature termination codon. Colicin M is regarded as the smallest of the colicins and is functionally unique, as it is the only colicin to inhibit peptidoglycan synthesis [24]. The *cma* gene containing a premature termination codon was annotated as two putative gene products: The first (N-terminal) fragment had low *aa* identity (29.0%) to the *cma* prototype UniProtKB: P05820, whereas the second (C-terminal) had 94.1% *aa* identity. The cognate *cmi* immunity genes in these isolates were maintained but retained 94.9% *aa* identity to the *cmi* prototype (UniProtKB: P18002) vs. 99.1–100% *aa* identity when these isolates harbored full-length *cma* genes. Thus, the functionality of either *cba* or *cma* in isolates 0638J and 0315J is not clear. Furthermore, although colicin B was always co-located with colicin M in this collection, colicin M was found without colicin B in one instance and was instead contiguous with colicin V (*cvaC*; truncated). 

#### 2.2.2. Colicin S4

Colicin S4 is a Group A colicin, which was detected in isolate 0315J harbored on a contig with identity (99% nucleotide identity; 85.1% query coverage) to *E. coli* strain G1/2 plasmid pSYM12 (a component of the human probiotic product Symbioflor 2 [25]; GenBank: KM107848). Colicin S4 (encoded by *csa*) is accompanied by both an immunity factor (*csi*) and a lysis protein (*csl*). In isolate 0315J, *csa* shares 99.2% *aa* identity with the known *csa* sequence (UniProtKB: Q9XB47), and the immunity and lysis proteins share 98.9% and 100% *aa* identity, respectively, to known sequences for *csa* and *csl* (UniProtKB: Q9XB46 and Q9XB46).

#### 2.2.3. Colicin R

The sole colicin gene in isolate 0842J was initially annotated as *cba*, as the C-terminal (142 *aa*) of *cra* shared 68.3% *aa* identity with the equivalent sequence in *cba*. However, the 0842J colicin gene had 99.8% *aa* identity to the colicin R activity gene *cra* (UniProtKB: T2D1N2), suggesting the colicin in 0842J and its cognate immunity and lysis proteins–*cri* and *crl*–comprise a variant of colicin R, a ‘group A’ Tol-dependent colicin. Colicin R was recently described and was found to be produced in biofilms formed by natural *E. coli* ROAR029 and exhibited increased activity against biofilm bacteria [26].

#### 2.2.4. Colicins Ia and Ib

The Group B colicin Ia and its associated immunity protein are encoded by *cia* and *iia*, respectively. Colicin Ib and immunity protein are encoded by *cib* and *iib*. Although the gene content and synteny of the colicinogenic loci in 0114J, 0096I, 0430J, and 0453J were similar, full-length variants of colicin Ia were found in isolates 0114J and 0096I, and variants of colicin Ib were found in 0430J and 0453J. The *cib* genes found in 0430J and 0453J were substantially different than *cia*, and shared only 80.8–81.3% *aa* identity to colicin Ia. A truncated remnant of *cia* which lacked *iia* was found in isolate 0043M, strongly suggesting colicin Ia in 0043M was nonfunctional. In isolates 0114J and 0096I, the *cia* and *iia* genes shared 99.0–99.3 % *aa* identity and 99.1% *aa* identity to the colicin Ia and immunity gene prototypes (UniProtKB: P06716 and Q46741, respectively). The colicin Ib genes in 0430J and 0453J each shared 98.7–98.8% *aa* identity with the colicin Ib prototype (UniProtKB: P04479). Likewise, the immunity genes, designated *iib*, shared 94.8–97.4% *aa* identity to the known colicin Ib immunity protein (UniProtKB: H9XP55). Interestingly, the *cib* immunity genes in 0430J and 0453J shared only 22.1% *aa* identity to *iia*. For both colicin Ia and Ib, all contigs identified matched various *E. coli* plasmids, and the contig harboring colicin Ib in isolate 0430J exhibited the highest identity (99.9% nucleotide identity; 70.5% query coverage) to a plasmid found in a ground turkey *S. enterica* isolate (GenBank: CP022064.2). The 0430J contig containing colicin Ib was notable for the presence of the ESBL *bla*_CMY-2_, and *sugE*, a determinant thought to confer resistance to quaternary ammonium compounds. 

#### 2.2.5. Colicins E1 and E2

Colicin E1 and E2 are Group A colicins and require the products of three genes sometimes known as *cea* (colicin E activity), *imm* (immunity), and *lys* (lysis protein). Here, we found variants of colicin E1 in isolates 0067K and 0143I on contigs with greatest identity to plasmids pCOLE1-H22 (from a 1978 Brazilian *E. coli* isolate [27]) and pEC276_KPC (GenBank: CP018949.1). The sequence of colicin E1 (and the associated immunity protein) found in isolate 0067K deviated (86% and 92.9% *aa* identity, respectively) from the prototype *cea* and *imm* sequences (UniProtKB: PP02978 and P02985). To the best of our knowledge, *cea* in 0067K encodes a novel ‘Colicin E’, although similar sequences are found in the NCBI *nr* database annotated as ‘colicin 10’ or ‘colicin E1’, despite lacking identity with the known sequence for colicin 10 (UniProtKB: Q47125) or colicin E1 (UniProtKB: P02978). The other colicin E1 detected in isolate 0143I was a near-identical variant (99.8% *aa* identity) to the colicin E1 prototype. In isolate 0089K, colicin E2 was found to be truncated and lacking the *lys* gene. However, the colicin E2 contig also encoded microcin J25 and the ESBL *bla*_CTX-M-55_, known to confer resistance to ceftazidime [28].

#### 2.2.6. Colicin V (Microcin)

Colicin V was detected in isolates 0143I, 0114J, 0638J, 0315J, and 0089K. The synthesis of colicin V involves four genes: *cvaA*, *cvaB* (a.k.a. *apxlB*, *ltxB*, or *mchF*), *cvaC* (the colicin V activity gene, 103 *aa* primary translation product), and the immunity gene, *cvi* [29]. We found *cvaC* was identical to the prototype *cvaC* (UniProtKB: P18002) in each isolate except for 0143I (72% aa identity). Isolate 0143I lacked *cvaA* and *cvaB* elsewhere in the whole-genome sequence and was the only colicin V-containing isolate in which *cvi* was not identical to the known immunity gene sequence. Taken together, this suggest colicin V in 0143I is nonfunctional. In isolate 0114J, colicin V was co-located with colicin 1A on a contig aligning (99.2% pairwise identity; 54.5% query coverage) with pCOV8, a colicin V-containing plasmid harbored by a commensal ESBL-producing *E. coli* isolated from the caecum of a broiler chicken in France [30]. The colicin V machinery in isolate 0315J was encoded on a contig aligning (100% pairwise identity; 100% query coverage) to pCVM29188_146, a 146,811 bp plasmid encoding streptomycin, and tetracycline resistance genes. Plasmid pCVM29188_146 was originally found in a ceftiofur-resistant S. *enterica* subsp. enterica serovar Kentucky isolated from chicken meat. Interestingly, the ~106kb plasmid backbone of pCVM29188_146 is known to share >90% nucleotide identity with two colicinogenic virulence plasmids from avian pathogenic *E. coli* (APEC) strains: pAPEC-O1-ColBM (encoding colicins B and M) and pAPEC-O2-ColV (encoding colicin V) [31].

#### 2.2.7. Microcin J25

Microcin J25 is synthesized as a 58 *aa* precursor peptide encoded by *mjcA*. The mature microcin (21 *aa*) has a ‘lasso’ structure (an 8 *aa* cyclic portion and 13-residue linear segment that loops back through the cyclic segment) [32]. This structure inhibits transcription by directly obstructing nucleoside triphosphate (NTP) from entering RNA polymerase, acting as a ‘cork in a bottle’ [32]. The microcin is divergently transcribed with an operon containing genes (*mjcB*, *mjcC*) involved in the maturation of the peptide, and *mjcD*, which serves the dual role of immunity and export [33]. Here, the microcin J25 cassette was detected in isolate 0089K on a contig bearing the greatest resemblance (99.7% nucleotide identity; 60.5% query coverage) to a large (>150 kbp) conjugative plasmid (GenBank: CP021198.1) from a clinical specimen isolated in China. In isolate 0089K, microcin J25 was co-located with a truncated remnant of colicin E2, which was likely nonfunctional.

#### 2.2.8. Microcin H47

Microcin H47 and its associated genes were found in a single isolate, 0067K. Microcin H47 is encoded in a structure designated the ‘microcin H47 small genomic island’, a chromosomal ~3 kb genetic system flanked by imperfect direct repeats. The microcin H47 small genomic island encodes all genes required for synthesis, post-translational modification, secretion, and immunity [34]. In strain 0067K, the entirety of the island was present with 93.9% nucleotide identity to the known sequence (GenBank: AJ009631.3). Microcin H47 is encoded by *mchB*, encoding a 75 *aa* peptide. In isolate 0067K, *mchB* shared 100% *aa* identity with the prototype (UniProtKB: P62530), whereas the immunity protein, encoded by *mchI*, shared 95.7% *aa* identity with its prototype (UniProtKB: O86200). The small genomic island also harbors genes for microcin I47 and the cognate I47 immunity protein (*mchS2* and *mchS3*, respectively). Microcin I47 has been found to be active under iron-limited conditions [34]. Consistent with the literature [34], the microcin H47 small genomic island in isolate 0067K was located on a contig with high identity to chromosomal sequences, indicating it was not plasmid-borne.

### 2.3. Activity against Shiga Toxin-Producing E. coli and Other Enteric Pathogens

To assess the antimicrobial activity of each of the putative bacteriocin-producing strains identified in the initial screen, we further tested each isolate against an in-house collection of STEC (n = 8) and other enteric pathogens (n = 20) in the absence of mitomycin C. Isolate 0430J (colicin Ib) consistently inhibited multiple human- and bovine-derived isolates of STEC O157:H7, other STEC serotypes (O111, O121, O-26, O-45, O145, O103, and O178), and certain Yersinia *spp*. and Salmonella *spp.* (Figure 1A). Similarly, isolates 0114J (colicin V and colicin Ia) and 0096I (colicin Ia) were found to inhibit multiple *E. coli* STEC isolates and Salmonella *spp.*, but not Yersinia *spp*. Overall, inhibition halos were not observed among test strains for most of the bacteriocin-producing *E. coli* identified. This suggests either the strains tested are resistant to most of the colicins and microcins tested, or these bacteriocins are not actively synthesized in the absence of induction with mitomycin C. Furthermore, none of the putative bacteriocin-producers inhibited other species tested, including *Acinetobacter baumannii, Bacillus cereus, Pseudomonas aeruginosa*, *Staphylococcus aureus*, *Enterococcus faecium*, *Klebsiella pneumonia*, and *Campylobacter jejuni* (data not shown).

### 2.4. Activity of Bacteriocin-Producing E. coli against Antimicrobial Resistant Isolates from Bovines and Wastewater

Each isolate in the *E. coli* collection was tested against every other isolate to assess their inhibitory activity against AMR *E. coli* in the absence of mitomycin C. This was done to identify additional putative bacteriocin-producing isolates. However, no additional putative producers were identified. Isolates 0430J, 0114J, 0096I, and 0089K were found to inhibit multiple isolates from the *E. coli* collection (Figure 1B), as did other putative bacteriocin-producing isolates (e.g., 0842J, 0043M, 0067K, and 0143I). Certain putative bacteriocin-producers were shown to be inhibited by other putative producers, but always encoded a different colicin and immunity protein. For example, isolates 0453J and 0430J, both harboring colicin Ib, were inhibited by isolate 0114J (colicins Ia and V). Isolate 0842K, harboring colicin R, was inhibited by producer isolates 0143I, 0114J, and 0089K, did not inhibit any of the known bacteriocin-producing *E. coli*, but otherwise inhibited ‘non-producer’ isolates 0008F, 008M, 0053J, 0143K, and 0638J. Of the 90 isolates, only 9 were not inhibited by any other isolate. To determine if the AMR phenotype was correlated with resistance to putative bacteriocin producer, Spearman’s rank-order was performed and identified no significant correlations between the AMR resistance profile and resistance to any of the bacteriocin-producing strains. 

### 2.5. Effectiveness of Bacteriocin-Producers in Broth-Based Competitions with E. coli K-12 Strain MG1655 or the Shiga Toxin-Producing E. coli Strain EDL933

Many of the putative bacteriocin-producers identified here did not exhibit activity against multiple *E. coli* under the conditions of the agar-based inhibition halo assay. Although isolates were treated with mitomycin C in the initial screen for putative bacteriocin-producing bacteria [4], mitomycin C was not used competition assays because one of the objectives of this work was to identify putative bacteriocin producers capable of inhibiting AMR *E. coli* without induction (i.e., constitutively expressed/secreted bacteriocins). To assess how putative bacteriocin producers might act in liquid culture in a competitive growth curve, luciferase-expressing *E. coli* K-12 strain MG1655 and O157:H7 strain EDL933 (pAKux2; ampicillin-resistant) were competed with the putative bacteriocin-producers, with light production monitored continuously for 12 h (Figure 4A). An OD-based growth curve of each putative bacteriocin-producing strain and the luciferase-expressing strains was taken to ensure each strain had comparable growth rates (Figure 4B). Likewise, CFU were enumerated at 0 hr and 12 hr on agar supplemented with ampicillin to assess viability of MG1655/pAKlux2 or EDL933/pAKlux2, following co-culture with the bacteriocin-producer isolates (Figure 4C). 

Via these methods, the isolates harboring bacteriocin genes decreased light production from MG1655/pAKlux2 co-cultures, lowering both the maximum detectable light production throughout the time course and at the endpoint (12 h) (Figure 4A, upper panel), an inhibitory response that corresponded to the agar-based method. The control for these experiments was an equivalent co-culture of MG1655 with MG1655/pAKlux2. All bacteriocin-producers prevented growth of MG1655/pAKlux2 to lower levels compared to the control co-culture or resulted in log-fold decreases in MG1655/pAKlux2 viability (Figure 4C). Likewise, all putative bacteriocin-producers exhibited inhibition of EDL933/pAKlux2 (Figure 4A, lower panel), contradictory to results with the agar-based assay (Figure 1A) This trend was generally reflected in the viable CFU counts (Figure 4C), in which most of the putative bacteriocin-producers, excluding isolates 0842J, 0043M, and 0067K, caused log-fold decreases in viability of strain EdL933. All putative bacteriocin-producers inhibited growth of EDL933/pAKlux2 as compared to the EDL933 and EDL933/pAKlux2 co-culture control. These results suggest putative bacteriocin-producers could be more effective against susceptible *E. coli* under specific conditions. These data also demonstrated that co-culture with luciferase-expressing *E. coli* may be a more rapid methodology for screening larger isolate libraries for antimicrobial activity.

## 3. Discussion

In this investigation, we performed an initial screen to identify *E. coli* isolates capable of producing diffusible substances inhibiting generic and AMR *E. coli*, STEC, and other enteric pathogens, and identified a variety of known genes coding for bacteriocins through whole-genome sequencing. One isolate potentially harbored up to four bacteriocin genes, a finding in agreement with research suggesting *E. coli* encoding more than one type of bacteriocin are relatively prevalent [4]. An obvious advantage of producing multiple bacteriocins is a wider spectrum of activity against competing bacteria, a wider receptor repertoire, and theoretically, enhanced fitness in more environments [35]. Putative bacteriocin-producers were found to be capable of inhibiting AMR bacteria and some producers inhibited numerous *E. coli*. However, our data highlighted some of the issues confounding both the identification and the effective application of bacteriocin-producers as alternative antimicrobials. These include: (i) Most putative bacteriocin producers identified here did not inhibit *E. coli* under the conditions of the agar-based assay, despite producing diffusible substances with antimicrobial activity under mitomycin C induction; (ii) in some instances, putative bacteriocin-producers with no apparent antimicrobial activity in the agar-based assay exhibited activity in the liquid luciferase-based assay; (iii) although some of the isolates produced log-scale decreases in viability in the target *E. coli*, none completely eliminated either generic *E. coli* K12 or STEC EDL933 when in co-culture; and (iv) these methods do not exclude other factors contributing in-whole or in-part to inhibition in co-culture assays. Other confounding variables include non-expression or incomplete maturation of the putative bacteriocins in different growth phases and culture conditions, and similar changes affecting susceptibility of the target population [25]. Putative bacteriocin-producers are likely to harbor other competitive mechanisms responsible for discrepancies between solid and liquid-based assays (e.g., contact-dependent growth inhibition [5]). Furthermore, the ecology of natural environments is more complicated than a two-strain competition. Although targeted bacteria may be initially reduced in natural environments, susceptible populations are likely to undergo post-exposure recovery. Constant exposure to bacteriocin-producers may overcome this limitation, provided the strain is fit and remains viable in the environment of interest. Based on current understanding of how certain colicins and microcins are produced and regulated, it is likely these issues will persist in isolates selected for development as probiotics. 

Some putative bacteriocinogenic isolates identified here were notable for phenotypic resistance to up to nine antimicrobials. Although we do not envision these particular isolates being directly developed as probiotics, the fact remains that bacteriocin-producers are not suitable probiotics if they harbor AMR genes or other virulence determinants. Although bacteriocins may be desirable features of probiotics, bacteriocin production is also a desirable trait for pathogenic bacteria [16]. 

Given that most of the bacteriocins identified in this study are hosted on self-transmissible plasmids, it would seem to be folly to use them against AMR bacteria if those same bacteria could easily acquire the bacteriocin and its cognate immunity gene. This would simply be recapitulating the problem of increasing AMR. This suggests purification (i.e., divorcing the bacteriocin from its immunity gene) is the way forward with bacteriocin development, a much more complicated endeavor than live probiotic administration. Work with purified microcin J25 has been found to be effective in a mouse model of infection, reducing viable counts of Salmonella Newport in the spleen and liver by 2–3 logs compared to peptide-free controls [36]. Likewise, dietary supplementation of microcin J25 improved performance and attenuated diarrhea in weaned pigs [37]. This demonstrates the therapeutic potential of bacteriocins. Therefore, it is important to continue identifying novel bacteriocins or variants for potential downstream development. Future work using strains/plasmids constructed with each of the bacteriocin-encoding loci independently is required to elucidate the effects of each of the putative bacteriocins identified here. Another drawback of the work presented here is that in silico identification generally relies on prior knowledge of bacteriocin sequences. As a result, our study did not identify novel bacteriocins. Furthermore, we only sequenced putative bacteriocin-producing isolates here, but understanding why certain isolates are resistant to putative bacteriocin producers is important if bacteriocins are to be useful as probiotics. Such strains might harbor bacteriocin immunity genes without the cognate bacteriocin, as we observed here in several instances, or other resistance mechanisms, such as outer membrane modifications [23].

Collectively, the results here demonstrate the complexity associated with identifying *bona fide* bacteriocin production and activity using putative bacteriocin-producers and susceptible *E. coli*. Although we initially used mitomycin C to induce bacteriocin production, it is not feasible to use mitomycin C in most downstream applications, such as in conjunction with probiotic administration to promote bacteriocin production in vivo. Consequently, we chose not to use mitomycin C when screening for putative bacteriocin-producers, hoping to identify isolates constitutively exerting antimicrobial activity. This putatively identified isolates 0096I and 0430J as meeting this criterion, which harbored variants of colicin Ia and colicin Ib, respectively. This result was confounded by other isolates, which contained colicin Ia and colicin Ib (and other bacteriocins) and did not exhibit the same activity between the agar-based assay and the luciferase-based assay. Furthermore, some isolates possessed the same colicin, but exhibited varying activity, as was the case with 430J and 453J, where both possessed colicin Ib, but only 430J inhibited multiple *E. coli* strains. Some possible explanations for this discrepancy include: (i) Variations in the colicin *aa* sequence confer different antimicrobial activity or different receptor binding and uptake capabilities; (ii) producer-specific peculiarities alter colicin production; and (iii) unknown defense mechanisms. Some evidence here suggests the latter might be more likely. For example, isolate 0453J, putatively produced colicin Ib that exhibited no activity against EDL933 in the agar-based assay, yet was effective in liquid competition, and inhibited EDL933/pAKlux2 in the luciferase-based assay, decreasing viability as demonstrated by a decline in CFU. Collectively, these data suggest luciferase-expressing *E. coli* used in competitive assays could be a rapid, but limited methodology for detecting putative bacteriocinogenic isolates and antimicrobial activity. Such luciferase-based competition screens have been deployed elsewhere [38]. The value of this method is sensitivity, the ability to observe the kinetics of a co-culture competition, and most significantly, the ability to screen hundreds of isolates simultaneously. However, the broth- and luciferase-based systems have several disadvantages compared to the agar-based assay, including: (i) The inability to distinguish between inhibition due to bacteriophages; (ii) unknown effects of luciferase expression on cell viability; and (iii) the inability to distinguish inhibition due to diffusible substances vs. inhibition due to other factors. In conclusion, we identified a number of putative bacteriocinogenic isolates capable of inhibiting AMR *E. coli* and STEC with diffusible products and demonstrated the utility of a luciferase-based assay for future screens for bacteriocin-producers.

## 4. Materials and Methods

### 4.1. Enrichment and Isolation of E. coli

*E. coli* were isolated from bovine feces, beef cattle feedlot catchbasins, and community wastewater in Alberta, Canada as similarly described [39]. Briefly, putative *E. coli* were enriched in Nutrient Broth (NB) supplemented with 2 µg/mL cefotaxime (MilliporeSigma, Etobicoke, ON, Canada) (grown overnight at 37°C shaking). Enrichments were plated for isolation on MacConkey agar containing 4 µg/mL cefoxitin (MilliporeSigma) and lactose-fermenting colonies were further isolated on tryptic soy agar (TSA) supplemented with 4 µg/mL cefoxitin. All media were BD Difco or BBL products (ThermoFisher, Burlington, ON, Canada). *E. coli* were confirmed with 16S rRNA sequencing and for indole production (Kovacs reagent; Fisher Scientific, Pittsburgh, PA, USA) as previously described.

### 4.2. Screening for Bacteriocin Production and Activity against Enteric Bacteria

A library of 90 confirmed *E. coli* was arrayed by inoculation into 96-well microplates containing 200 μL of Mueller-Hinton (MH) II broth. MH was chosen because of its routine use in antimicrobial susceptibility testing. Bacterial growth at 37 °C was monitored by spectrophotometry (BioTek HTX Synergy plate reader) to an OD_600nm_ of ~0.2 then diluted 1/100 into fresh MH broth containing 0.2 μg/mL mitomycin C (MilliporeSigma) in a new 96-well plate and incubated for 1 h. To test for activity against non-pathogenic or Shiga toxin-producing *E. coli* (STEC), 5 μL from each well was spotted onto a single-well microplate containing MH agar recently inoculated either with 0.005 OD_600nm_ of *E. coli* K-12 (strain MG1655) or *E. coli* O157:H7 strain EDL933, a well-characterized enterohaemorrhagic STEC. Plates were incubated at 37°C for 18 h and then imaged using the BIOMIC V3 Microbiology System (Giles Scientific). Inhibition halos in underlying *E. coli* lawns were measured with ImageJ (NIH). The arrayed library was likewise tested for activity against each isolate present in the library to further identify bacteriocin-producing *E. coli*, and to assess their capacity to inhibit *E. coli* with known antimicrobial susceptibility profiles. Isolates capable of inhibiting any *E. coli* strain were selected for additional testing against an in-house collection of STEC of human or bovine-origin (O157, O26, O45, O103, O111, O121, O145, O178) [40] and other bacterial species, including *Acinetobacter baumannii* (ATCC 17978), *Bacillus cereus* (ATCC 14579), *Campylobacter jejuni* (ATCC 33560), *Clostridium difficile* (ATCC 9689), *Enterococcus faecium* (ATCC 19434), *Klebsiella pneumonia* (ATCC 700603), *Klebsiella pneumoniae* (ATCC 4352), *Listeria monocytogenes* (ATCC 19117), *Pseudomonas aeruginosa* (ATCC 27853), *Salmonella enterica* (Braenderup), *Salmonella enterica* Enteritidis (ATCC 13076), *Salmonella enterica paratyphi* (ATCC 9150), *Salmonella typhimuriam* (ATCC 13311), *Staphylococcus aureus* (ATCC 25923), *Staphylococcus aureus* (ATCC 29213), *Staphylococcus aureus* (ATCC 29740), *Staphylococcus aureus* (ATCC 35556), *Streptococcus pneumonia* (ATCC 33400), *Yersinia enterocolitica* (ATCC 9610), and *Yersinia pseudotuberculosis* (ATCC 6904). In brief, bacterial lawns were swabbed on MH agar from a bacterial suspension standardized to ~ OD_600nm_ 0.1 in sterile saline. Next, 5 μL (OD_600nm_ 0.005) of bacteriocin-producers was spotted onto the agar surface, and inhibition halos were measured following growth for 20 h at 37 °C. Bacteriophage vs. bacteriocin production was differentiated as described elsewhere [41]. Essentially, inhibitory activity due to diffusible substances was assessed by serial dilution of supernatants from cultures of putative bacteriocin-producers on lawns of *E. coli* K-12 and confirmed negative for the presence of phage plaques.

### 4.3. Antimicrobial Susceptibility Assays

Disc-based susceptibility testing was performed using CLSI guidelines (CLSI document M02-A12 and CLSI supplement M100S) [42,43]. *E. coli* isolates were tested for resistance to oxytetracycline, trimethoprim/sulfamethoxazole, ampicillin, florfenicol (Oxoid), neomycin, sulfisoxazole, streptomycin, ceftiofur (Oxoid), ceftazidime, and amoxicillin/clavulanate. Unless otherwise indicated, all tests were conducted with BD BBL Sensi-Disc antimicrobial susceptibility test discs (BD). Zones of inhibition were measured using the BioMic V3 imaging system (Giles Scientific, Santa Barbara, CA, USA). Except for neomycin, which used EUCAST criteria (www.eucast.org), CLSI criteria [42,43] was used to categorize isolates as ‘sensitive’ or ‘resistant’. Here, ‘intermediate’ resistance was designated as ‘sensitive’.

### 4.4. Broth-Based Co-culture and Activity Assay with Luciferase-Expressing E. coli

*E. coli* K-12 strain MG1655 and *E. coli* O157:H7 strain EDL933 were electroporated with the luciferase-expressing plasmid pAKlux2 [44] (ampicillin-resistant) and recovered on LB agar supplemented with 100 μg/mL ampicillin. Light production was confirmed in selected colonies using a FluorChem HD2 (Alpha Innotech). To assess inhibition in broth culture, light production was measured repeatedly in OD-equivalent 200 μL MH broth (without antimicrobials) co-cultures of either MG1655/pAKlux2 or EDL933/pAKlux2 (at OD_600nm_ 0.005) and bacteriocin-producers (OD_600nm_ 0.005; total initial OD_600nm_ 0.01 in all tests and growth curves) in black clear-bottom 96-well plates (Nunc) using luminometry (BioTek HTX Synergy plate reader). Inhibition was registered as decreased light emission compared to control co-cultures, where either MG1655 or EDL933 (without pAKlux2) were equivalently inoculated with the luciferase-expressing version. For CFU-based co-culture experiments, CFU were enumerated on MH supplemented with 100 µg/mL of ampicillin.

### 4.5. Whole-Genome Sequencing and Bioinformatic Identification of Bacteriocin Genes

To characterize putative bacteriocin-producers, select isolates (those inhibiting MG1655 or EDL933 under mitomycin C exposure) were whole-genome sequenced. Briefly, DNA was extracted (DNeasy Blood and Tissue kit, Qiagen, Germantown, MD, USA) from *E. coli* cultured on BHI agar plates and prepared for Illumina MiSeq 2 × 300 paired-end sequencing at the Canadian Food Inspection Agency Lethbridge Laboratory. Trimmomatic 0.38 was used to trim reads and the Illumina adaptors with criteria: phred33, LEADING:3, TRAILING:3, SLIDINGWINDOW:4:15, MINLEN:36. SPAdes 3.13.0. [45] and PROKKA [46] were used for contig assembly and annotation, respectively. To identify bacteriocins in silico, contigs were searched for hits aligning (blastx in Geneious 8.1.9) with 230 bacteriocins downloaded from BACTIBASE [47]. Hits with >30% pairwise *aa* identity were retained and scrutinized manually for co-localization with other bacteriocins and immunity protein homologues, which were further identified in adjacent ORFs using NCBI CD search (NIH) [48]. Sequences from putative bacteriocin-producers were assessed with BAGEL3 for comparison [49].

### 4.6. Data Visualization and Statistical Analyses

For pairwise sequence comparisons, bacteriocin and immunity genes were translated (bacterial transl_table 11) and aligned with MUSCLE (default parameters) in Geneious 8.1.9. Interactions between bacteriocin-producers and susceptible *E. coli* were visualized with Circos [50]. Gene diagrams and alignment identities were produced in EasyFig [51]. All statistical tests were one-way ANOVAs (multiple comparisons vs. control group; Bonferroni t-test) performed in Sigmaplot 13.0 (Systat Software Inc.). *P*-value summary: Not significant (*P* > 0.05); * (*P* ≤ 0.05); ** (*P* ≤ 0.01); *** (*P* ≤ 0.001). Error bars indicate standard error of the mean (SEM).

### 4.7. Nucleotide Sequence Accession Numbers

Bacteriocin-containing contig sequences were deposited in GenBank under accession numbers MK878515 to MK878535. Illumina sequence data were deposited in the NCBI Short Read Archive under BioProject ID PRJNA556083. 

## Figures and Tables

**Figure 1 toxins-11-00475-f001:**
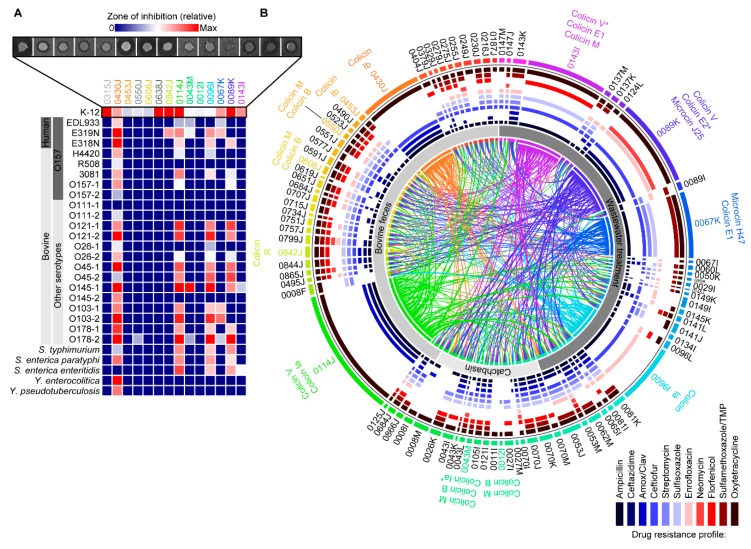
Activity of putative bacteriocin-producing *E. coli* against antimicrobial-resistant (AMR) *E. coli*, serotyped *E. coli*, and other enteric bacteria. (**A**) Zones of inhibition and an activity heatmap of putative bacteriocin-producing *E. coli* against human- and bovine-derived shiga toxin-producing *E. coli* (STEC), including O157:H7 and other STEC serotypes associated with food-borne disease and other enteric bacteria. (**B**) Circular plot showing activity against *E. coli* isolates from bovine feces, feedlot catchbasins, and wastewater influent. Bacteriocin-producers are labelled, and antimicrobial activity is indicated by a link directly terminating at the colored bar representing the susceptible isolate. The AMR profile is shown for each isolate. The width of each link indicates the relative inhibition halo produced by each bacteriocin-producing strain. Isolates are not shown if they neither inhibited nor were inhibited by any other *E. coli*.

**Figure 2 toxins-11-00475-f002:**
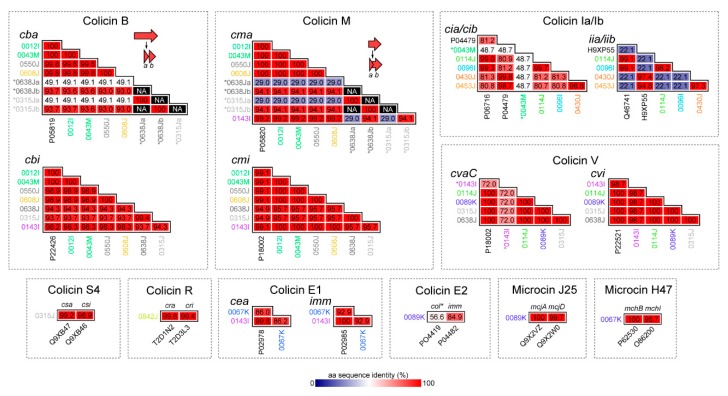
Summary of amino acid (*aa*) identity of putative bacteriocins and cognate immunity genes identified in silico. Truncated genes are indicated with asterisks or shown as multiple arrows.

**Figure 3 toxins-11-00475-f003:**
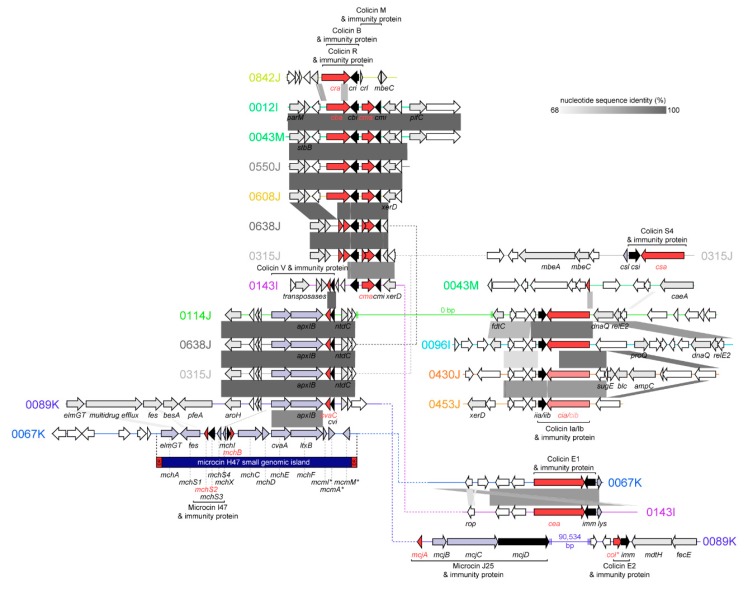
Linear comparisons of genomic loci encoding bacteriocins identified in silico in the genomes of putative bacteriocin-producing *E. coli* isolates. Amino acid (*aa*) sequence identity to prototype bacteriocins is shown. Truncated genes are indicated with asterisks or shown as multiple arrows. Potential genes are depicted as arrows showing bacteriocin genes (red), immunity genes (black), colicin-associated genes (light-blue), genes of known function (grey, labelled), and conserved hypothetical proteins (white). Solid lines (and *bp* distance) indicate bacteriocin gene clusters found on the same contig; dashed lines indicate different contigs.

**Figure 4 toxins-11-00475-f004:**
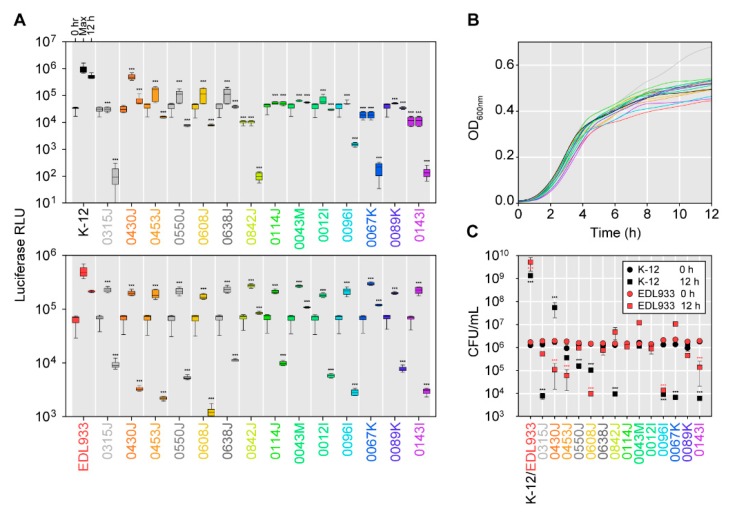
Co-culture broth-based competitions between bacteriocin-producing isolates and indicator luciferase-expressing *E. coli* K-12 strain MG1655/pAKlux2 and *E. coli* O157:H7 strain EDL933/pAKlux2. (**A**) Upper panel: OD-equivalent co-culture of putative bacteriocin-producers or *E. coli* MG1655 with MG1655/pAKlux2. Lower panel: Co-culture with EDL933/pAKlux2. Data shown are log-scale luminometric relative light units (RLU) with initial, maximal, and endpoint measurements shown for a 12 h time course. Mean of 12 biological replicates; error bars indicate SEM. (**B**) Growth-curves (12 h; OD_600nm_) of bacteriocin-producing isolate monocultures. No significant differences in growth rate were observed between isolates and *E. coli* MG1655/pAKlux2 or EDL933/pAKlux2. (**C**) CFU (ampicillin-resistant) recovered from co-culture competitions at 0 h and 12 h timepoints. Data shown are the mean of three biological replicates; error bars indicate SEM. Statistical tests are one-way ANOVAs; *P*-value summary: * (*P* ≤ 0.05); ** (*P* ≤ 0.01); *** (*P* ≤ 0.001).

**Table 1 toxins-11-00475-t001:** Summary of bacteriocins identified in sequenced *E. coli* isolates.

*E. coli* Isolate	Isolation Source	AMR Profile^1^	AMR Genes^2^	Bacteriocins Identified	Contig BLAST Hit^3^	Accession Numbers
0315J	Bovine feces	AMP, AMXC, STR, OXYT	^V^ *macAB*	colicin M * colicin B * colicin S4 colicin V	^M,B^*S. enterica* Saintpaul pCFSAN004174 ^S4^pSYM12 ^V^*S. enterica* Kentucky pCVM29188_146	^M,B^CP019207 ^S4^KM107848 ^V^CP001122
0430J	Bovine feces	AMP, AMXC, CTZD, CTIO, OXYT	^Ib^ *bla* _CMY-2_ ^Ib^ *sugE*	colicin Ib	^Ib^*S. enterica* pFDAARGOS-312-4	^Ib^CP022064
0453J	Bovine feces	AMP, CTIO, SULF, TMSZ, OXYT		colicin Ib	^Ib^p2HS-C-2	^1A^CP038182
0550J	Bovine feces	AMP, CTIO, STR, SULF, ENRO, FLOR, OXYT		colicin M colicin B	^M,B^FHI82 plasmid contig	^M,B^LM996773
0608J	Bovine feces	AMP, CTZD, CTIO, STR, SULF, ENRO, FLOR, TMSZ, OXYT		colicin M colicin B	^M,B^p2013C-4404	^M,B^CP027378
0638J	Bovine feces	AMP, AMXC, STR, SULF, OXYT		colicin M * colicin B * colicin V	^M,B^p2013C-4404-2 ^V^pDSM30083	^M,B^CP027378 ^V^CP033091
0842J	Bovine feces	AMP, AMXC, CTZD, CTIO, STR, SULF, FLOR, TMSZ, OXYT		colicin R	^R^p14408-3	^R^LT599828
0114J	Bovine feces	AMP, AMXC, CTZD, FLOR, TMSZ, OXYT		colicin V colicin Ia	^V,Ia^pCOV8	^V,1A^MG648896
0043M	Feedlot catchbasin	AMP, AMXC, CTZD, CTIO, STR, SULF, FLOR, TMSZ, OXYT	^M,B^*aph*(6)-Id	colicin M, colicin B colicin Ia *	^M,B^*S. enterica* Heidelberg p12-4373-62 ^Ia^p2014C-3075	^M,B^CP012928 ^1A^CP027448
0012I	Feedlot catchbasin	STR, OXYT	*aph(6)-Id tet(C)*	colicin M colicin B	^M,B^pExPEC-XM	^M,B^CP025329
0096I	Wastewater influent	AMP, CTIO, ENRO		colicin Ia	^Ia^pLKSZ01	^1A^CP030282
0067K	Wastewater influent	AMP, CTIO, STR, SULF, TMSZ, OXYT		colicin E1 microcin H47	^E1^pCOLE1-H22 ^H47^NCTC10444	^E1^AY913943 ^H47^LR134092
0089K	Wastewater Influent	AMP, CTZD, CTIO, ENRO, NMYN, OXYT	^E2,J25^ *bmrA*	colicin V colicin E2 * microcin J25	^V^pAMSC2 ^E2,J25^pH17-5	^V^CP031107 ^E2,J25^CP021198
0143I	Wastewater influent	AMPI, CTIO, SULF, FLOR, OXYT	^M,V^ *bla* _CTX-M-55_	colicin E1 colicin M colicin V *	^E1^pEC276_KPC ^M,V^BR02-DEC	^E1^CP018949 ^M,V^CP035320

* Truncated (potentially defective). ^1^ Abbreviations: AMP, ampicillin; AMXC, amoxicillin-clavulanate; CTZD, ceftazidime; CTIO, ceftiofur; STR, streptomycin; SULF, sulfisoxazole; ENRO, enrofloxacin; NMYN, neomycin; FLOR, florfenicol; TMSZ, trimethoprim-sulfamethoxazole; OXYT, oxytetracycline. ^2^ Antimicrobial resistance genes present on bacteriocin contigs. ^3^
*E. coli* unless otherwise indicated; superscripts indicate bacteriocin and associated contig.

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
