# Peer review of "Bacteriocin Occurrence and Activity in Escherichia coli Isolated from Bovines and Wastewater"

_toxins, 2019, doi:10.3390/toxins11080475_

Round 1

Reviewer 1 Report

The manuscript was much improved and the authors addressed most of the points raised previously. Additional points are indicated below. In particular, the abstract would benefit from revision.

Comments about the abstract

L12. The terms « putative ESBL » is misleading for several reasons. (i) gold standard methodologies used worlwide (and recommended by CLSI or EUCAST) were not used ; (ii) phenotypic tests were not used (or not stated clearly) to confirm that these strains were indeed ESBL producers ; (iii) use of cefoxitin is indicative of AmpC producer (not ESBL).

L13. Only STEC and related enteric pathogens are mentioned here. Testing activity against AMR E. coli should also be included in this sentence.

L14.  The amount of 14% seems incorrect and the total number of strains screened should be stated here (e.g. « 14 (15,5%) out of 90 E. coli isolates tested…»)

L15. « identical, variant or truncated genes » : compared to what ?

L16-17. The use of the word « constitutive » is misleading at it suggests that upon induction (eg. using MMC) most of the bacteriocin-producers would be active against the E. coli strains, which either is not the case (eg. only 4 strains active against EDL933 in the presence of MMC, Fig 1A) or has not been tested.

L18. Please be more specific, eg. « with luciferase-expressing E. coli K12 or STEC EDL933».

L19. It is not clear what the authors mean by « yet to be fully realized »?

Other comments

L47. « also differ »

L92. Reference 18 is not about assaying fecal shedding in cattle.

L100 (and throughout the manuscript ; eg. 103, 129 (Fig.1), 307, 310, 448, 450, 458). As stated above the terms « ESBL-enriched » are misleading.

L215 : cib and iib ?

L256-258 and 272-273. Please specify that no MMC was used here as it is an important experimental condition that differentiates these new assays from the initial screen.

L292. Please specify how many STEC and enteric pathogens were tested.

L293-295. Strains 430J and 453J both carry colicin Ib gene but only 430J inhibits multiple strains. This point is intriguing and thus should be pointed out and further discussed.

L327. « in any other test » : please be more specific. 

L330. Please specify that this assay was performed in the absence of MMC.

L354. In the case of MG1655, results with the liquid assay seem similar to those with the agar-based method. This should be stated here.

L358. « viability of strain EDL933 ».

L367. Inhibiting.

L424. The initial screen (performed without MMC) is an exception that should be specified here for clarity.

L432. It is not clear why the existence of unknown defense mechanism is the most likely explanation for the discrepancies observed between the agar-based assay and the luciferase-based assay  since only two strains were used in both assays (ie K12 and EDL933) and  their defense mechanism should not differ from one assay to the other.

L.453 : If the authors wished to select for BLSE, why using cefoxitin in the agar plates (as cefoxitin is used for selecting AmpC producer, not BLSE) ?

L458. « 90 confirmed ESBL-enriched E. coli » : it is not clear which characteristic was confirmed : E.coli species, ESBL phenotype, or both ?

Author Response

Comments about the abstract

L12. The terms « putative ESBL » is misleading for several reasons. (i) gold standard methodologies used worlwide (and recommended by CLSI or EUCAST) were not used ; (ii) phenotypic tests were not used (or not stated clearly) to confirm that these strains were indeed ESBL producers ; (iii) use of cefoxitin is indicative of AmpC producer (not ESBL).

Response: As requested we have deleted the reference to ESBL from the manuscript.

L13. Only STEC and related enteric pathogens are mentioned here. Testing activity against AMR E. coli should also be included in this sentence.

Response: As requested we have added reference to activity against AMR E. coli into the abstract..

L14.  The amount of 14% seems incorrect and the total number of strains screened should be stated here (e.g. « 14 (15,5%) out of 90 E. coli isolates tested…»)

Response: As requested we have modified this to read as 14 out of 90 E. coli isolates tested,   Reader can calculate the percentage if they wish to do so.

L15. « identical, variant or truncated genes » : compared to what ?

Response: Sentence revised to “ Based on alignment within BACTIBASE, colicins M, B, R, Ia, Ib, S4, E1, E2, and microcins V, J25, and H47, encoded by identical, variant, or truncated genes were identified.”

L16-17. The use of the word « constitutive » is misleading at it suggests that upon induction (eg. using MMC) most of the bacteriocin-producers would be active against the E. coli strains, which either is not the case (eg. only 4 strains active against EDL933 in the presence of MMC, Fig 1A) or has not been tested.

Response: “Constitutive” deleted as requested.  

L18. Please be more specific, eg. « with luciferase-expressing E. coli K12 or STEC EDL933».

Response: Strain specific identifications have been added directly to the manuscript.

L19. It is not clear what the authors mean by « yet to be fully realized »?

 Response: Agreed this statement is cryptic – we have deleted this portion of the sentence.

Other comments

L47. « also differ »

Response: “also” added.

L92. Reference 18 is not about assaying fecal shedding in cattle.

Response: Thanks for picking this up the reviewer is correct. Sentence has been modified to:

“ Although probiotic colicinogenic E. coli strains have been found to exhibit activity against STEC in vitro [18], many probiotics have shown mixed results in reducing O157:H7 fecal shedding in cattle [19]”

L100 (and throughout the manuscript ; eg. 103, 129 (Fig.1), 307, 310, 448, 450, 458). As stated above the terms « ESBL-enriched » are misleading.

Response: As requested by the reviewer we have removed reference to ESBL from the manuscript. We have retained reference to the fact that the isolates were enriched in the presence of cefotaxime and selected for on cefoxitin. Although this does not meet the clinical definition of ESBL, it does make it much more likely that we would isolate ESBL’s using this procedure.

L215 : cib and iib ?

Response:   Corrected as requested.

L256-258 and 272-273. Please specify that no MMC was used here as it is an important experimental condition that differentiates these new assays from the initial screen.

Response:   We are not sure that the reviewer is referring to the correct line numbers here, but we have indicated that mitomycin C was not included in the competition assays.

L292. Please specify how many STEC and enteric pathogens were tested.

Response:   Corrected as requested (n=8 and n=20).

L293-295. Strains 430J and 453J both carry colicin Ib gene but only 430J inhibits multiple strains. This point is intriguing and thus should be pointed out and further discussed.

Response: Furthermore, some isolates possessed the same colicin, but exhibited varying activity as was the case with 430J and 453J, where both possessed colicin Ib, but only 430J inhibited multiple E. coli strains.

L327. « in any other test » : please be more specific. 

Response: Modified to “not used in competition assays”

L330. Please specify that this assay was performed in the absence of MMC.

Response: Indicated as requested.

L354. In the case of MG1655, results with the liquid assay seem similar to those with the agar-based method. This should be stated here.

Response: Modified as requested. “production throughout the time course and at the endpoint (12 h) (Figure 4A, upper panel), an inhibitory response that corresponded to the agar-based method.”

L358. « viability of strain EDL933 ».

Response: added as requested.

L367. Inhibiting.

Response: modified as requested.

L424. The initial screen (performed without MMC) is an exception that should be specified here for clarity.

Response: modified as requested.

L432. It is not clear why the existence of unknown defense mechanism is the most likely explanation for the discrepancies observed between the agar-based assay and the luciferase-based assay  since only two strains were used in both assays (ie K12 and EDL933) and  their defense mechanism should not differ from one assay to the other.

Response: We are not sure as to what the reviewer would like to see altered here. We do think that unknown defense mechanisms is still a possible explanation. It is also possible that these expression of these defense mechanisms differ between agar vs liquid culture. We have attempted to point this out further in the discussion..

L.453 : If the authors wished to select for BLSE, why using cefoxitin in the agar plates (as cefoxitin is used for selecting AmpC producer, not BLSE) ?

Response: We have removed the term putative ESBL producers throughout the mansucript. However, we do have other data that indicates that this enrichment process dramatically increases the likelihood of isolation of verified ESBL as well.

L458. « 90 confirmed ESBL-enriched E. coli » : it is not clear which characteristic was confirmed : E.coli species, ESBL phenotype, or both ?

Response: We have removed the term putative ESBL producers throughout the mansucript.

Reviewer 2 Report

The authors have corrected their manuscript according to reviewers' comments and have improved the presentation of their study

Author Response

We appreciate the reviewers time and input.

Reviewer 3 Report

The manuscript has been improved based on the reviewers' comments.

Author Response

(The authors gave the same response as above.)

Reviewer 4 Report

Interesting study. Very well written. Experiments appear to be well designed and executed. Page 3 , insert a space between "deemed" and "putative:. I suggest a new paragraph for the section on page 12:  "Given that  most of the bacteriocins identified in this study are hosted on self-transmissible plasmids, it would seem to be folly to use them against AMR bacteria if those same bacteria could easily acquire the bacteriocin and its cognate immunity gene. This would simply be recapitulating the problem of increasing AMR. This suggests purification (i.e., divorcing the bacteriocin from its immunity gene) is the way forward with bacteriocin development, a much more complicated endeavor than live probiotic administration "  ,because this is a VERY important point; a point that should also be mentioned in the abstract, so that the work is not misunderstood.

Author Response

Interesting study. Very well written. Experiments appear to be well designed and executed

Page 3 , insert a space between "deemed" and "putative

Response: Revision implemented.

:. I suggest a new paragraph for the section on page 12:  

"Given that  most of the bacteriocins identified in this study are hosted on self-transmissible plasmids, it would seem to be folly to use them against AMR bacteria if those same bacteria could easily acquire the bacteriocin and its cognate immunity gene. This would simply be recapitulating the problem of increasing AMR. This suggests purification (i.e., divorcing the bacteriocin from its immunity gene) is the way forward with bacteriocin development, a much more complicated endeavor than live probiotic administration”

Response: We have started this section as a new paragraph as requested.

because this is a VERY important point; a point that should also be mentioned in the abstract, so that the work is not misunderstood.

Response: We have also added a statement to the abstract as requested.

“These abundant toxins may have real potential as next-generation control strategies in livestock production systems, but divorcing the bacteriocin from its immunity gene maybe necessary for such a strategy to be effective.”

This manuscript is a resubmission of an earlier submission. The following is a list of the peer review reports and author responses from that submission.

Round 1

Reviewer 1 Report

This article describes the identification of bacteriocin-producing E. coli strains that target STEC and ESBL-producing strains. This topic is of particular interest in the context of increase of antimicrobial resistance and STEC-associated disease burden. A total of 14 strains were characterized by whole genome sequencing and gene clusters encoding putative bacteriocin were identified. The manuscript is well written although the presentation of the data would benefit from a new layout of the figures or inclusion of table (instead of a long description of the many genes identified and homologies found). The fact that discrepancies were observed between the different methods used (agar assays versus co-cultures) deserves further investigation. Moreover, mutation (or cloning into E. coli laboratory strain) of the identified bacteriocin-encoding genes would substantiate the conclusions made and confirm the role of these genes in the inhibition of STEC or  ESBL-producing strains. 

Major comments.

What is the rational behind the treatment of the library of E. coli strains with mitomycin C (MMC) when testing for antimicrobial activity ?  MMC is usually used to trigger the SOS response and induce prophages. Please clarify.

The method for differenciating between phages and bacteriocins (mentioned L463-464) is important here as MMC was used in the agar assay. This method should be described briefly in the Materials and Methods section, and the results obtained showing that STEC or ESBL inhibition was caused by bacteriocins (and not phages) should be described in the Results section. 

Figure 2 is too dense and should be modified for clarity. A figure dedicated only to a schematic representation of the bacteriocin gene clusters would be clearer (with homologies presented in a separated table or figure).

Other comments.

L133 : strain 315J is not highlighted as containing a truncated cba gene in Table 1.

L133-134 and 139 : It is not clear whether the truncations relate to genes (e.g. upon deletion) or to the encoded proteins (e.g. through the introduction of a stop codon).

L138 : two strains (638J and 315J) with a truncation in cma, contrasting with Table 1 (only 638J with a truncation in cma). 

L140 : « formed » => encoded. It is not clear if two proteins are really produced instead of one (resulting from a deletion in the cma gene).  There is no experimental evidence to support this.

L156-157 : to which group belongs colicin R ?

L157 : lone => sole

L246-247 : which « known sequence » the authors are refering to ?

L250 : « the island encoded genes » is incorrect.

L260-261 : strain 114 contains also colicin V encoding gene (not only Ia).

L276 : multiple isolates were inhibited: does they include both STEC and ESBL strains, or only one category ?

L274-278 : Why should there be an agreement between STEC and ESBL strains regarding their sensitivity to bacteriocins ? In addition, it is not clear why strain 430J is listed twice (L275 and L277). Strains 67K and 143 seem to target multiple strains too : why mentioning them in the second group ?

L284-286 : correlation between AMR profile and resistance to bacteriocin was examined. What is the relevance of this ? Phylogenetic analysis (e.g. using cgMLST or phylogrouping) would have been more informative regarding the genetic background and relatedness of the strains.

L295 : is mitomycin C a known inducer of the different types of bacteriocins identified here ? Please provide a reference.

L321-322 : « All bacteriocin-producers prevented growth of MG1655/pAKlux2 to levels observed in the control co-culture » : do the authors mean «  to lower levels compared to the control co-culture ? This sentence is not clear. 

L333-351 : the description of the context would be more appropriate in the introduction. 

L398-400 : It can not be excuded that other unknown bacteriocins were responsible for the inhibition observed. 

Reviewer 2 Report

 The authors of the present manuscript have studied the production of bacteriocins with anti-E. coli activity, and via whole-genome sequencing they have characterized the bacteriocin content of ESBL-positive E. coli strains isolated from bovine feces and wastewater.

The manuscript is very interesting, the applied methods are very well described and the manuscript very well written, however, minor points might be discussed.

a.       Paragraph 2.3: Did you test whether the strains that were resistant to the colisins and microsins tested carried any of the known colisin-immunity genes??? If not, please add a comment in the Discussion.

Reviewer 3 Report

The manuscript reports on an initial screen for E. coli isolates from bovine feces and from wastewater, which produce bacteriocins that inhibit ESBL-expressing E. coli, STEC and other enterobacteria. For this purpose, the authors treated a collection of 90 E. coli isolates with mitomycin C and tested their antibacterial activity against a lawn of E. coli K-12 strain MG1655 or STEC O157:H7 strain EDL933 on agar plates. 14 isolates were identified to produce diffusable compounds which inhibited growth of strain MG1655, and four of them also inhibited growth of STEC strain EDL933. The genomes of these 14  isolates were sequenced with Illumina technology. The draft genome sequences were then automatically annotated and in silico searched for bacteriocin determinants by BACTIBASE. As a result, at least one and up to four bacteriocin determinants were detected in the individual draft genomes. Colicin B and M determinants were most frequently detected (50 % of all bacteriocin producing strains), followed by colicin V (~36 % of all bacteriocin producers) and colicin Ia, Ib (in 14 % of all producers). Several other microcin and colicin determinants were less frequently detected. The individual bacteriocin determinants detected displayed marked sequence variability: not only identical bacteriocin determinants were detected, but also sequence variants as well as truncated genes.

Antimicrobial activity oft he 14 isolates was tested on agar against a selection of important STEC serotypes and different enterobacterial species as well as against different nosocomial pathogens incl. A. baumannii, B. cereus, P. aeruginosa, S. aureus, E. faecium, K. pneumoniae and C jejuni. Four isolates inhibited growth of multiple strains, whereas many others only interfered with growth of some other isolates. The antibiotic resistance profile had no effect on the bacteriocin susceptibility of the E. coli isolates.

The killing efficacy of the 14 bacteriocin producers in competition with E. coli MG1655 or E. coli EDL933 was tested in liquid broth in the absence of mitomycin C. For this purpose, light emission of luciferase-expressing variants of strain MG1655 and EDL933 was recorded for 12 hours during competitive growth with the individual bacteriocin producers. Additionally, the bacterial number of the two plasmid-bearing E. coli reporter strains was determined on ampicillin agar plates after 12 hours of competition. Based on the luciferase activity measurement, competition of the bacteriocin producer with the two reporter strains resulted in a sometimes as much as two-log reduction of light emission after 12 hours of competition. Similarly, also the number of viable bacteria decreased in this time span. Also in case of the STEC strain EDL933, a marked reduction of luciferase expression and cellular counts was observed, which was in contrast to the results obtained on agar plates.  All in all, the authors developed a luciferase-based test system for bacterial growth inhibition that may help to identify antagonistic factors expressed by bacteria. They report that the outcome of broth-based and agar plate-based growth inhibition assays may differ and that the broth-based assay may be more sensitive than the agar plate-based approach. They identified several strains which “constitutively“ express antagonistic factors that may interfere with growth of ESBL-producing or Shiga toxin-expressing E. coli variants. The authors discuss their findings against the background of urgently needed alternatives to antibiotics and the possibility to use purified bacteriocins to reduce colonization by (multi-)resistant or pathogenic E. coli. This is an interesting manuscript.

Points for consideration:

1)      One weak point of the study is the fact that there was no direct proof that killing of the reporter strains was indeed caused by bacteriocin expression. The authors discuss the limitations of their study and briefly touch this point in the Discussion.

2)      To what extent may bacteriocin expression affect luciferase expression of the bacteria without killing them?

3)      Was the broth-based competition assay performed in the presence of ampicillin? This is not quite clear from the text.

4)      How stable is the reporter plasmid in the presence of the bacteriocin producing strains? Can the authors exclude that the reduction in light emission and colony count of the two reporter strains may result at least in part from the mere loss of the reporter plasmid and not from bacteriocin-mediated killing?

5)      Did the authors check whether the EDL933-based reporter strain may kill or interfere with growth of some of the bacteriocin producers when grown in competition?

6)      Do the growth curves depicted in Figure 3b result from monocultures or from the competition experiments? Growth of the individual strains in monoculture may markedly differ from that in competition.